# Linking pollen deposition, snow accumulation and isotopic composition on the Alto dell'Ortles glacier (South Tyrol, Italy) for sub-seasonal dating of a firn temperate core

Daniela Festi[1], Luca Carturan[2], Werner Kofler[1], Giancarlo dalla Fontana[2], Fabrizio de Blasi[2], Federico Cazorzi[3], Edith Bucher[4], Volkmar Mair[5], Paolo Gabrielli[6,7], Klaus Oeggl[1]

[1]Institute of Botany, University of Innsbruck, Sternwartestraße 15, A-6020 Innsbruck, Austria

[2] Department of Land, Environment, Agriculture and Forestry, University of Padova, Viale dell'Università 16, 35020 Legnaro (PD) Italy

[3] Department of Agriculture and Environmental Sciences, University of Udine, via delle Scienze 208, 33100 Udine, Italy

[4] Autonome Provinz Bozen Südtirol, Landesagentur für Umwelt, Biologisches Labor, Unterbergstraße 2, 39055 Leifers, BZ, Italy

[5] Autonome Provinz Bozen Südtirol, Amt für Geologie und Baustoffprüfung, Eggentalerstraße 48, 39053 Kardaun (BZ), Italy

[6]Byrd Polar and Climate Research Center, The Ohio State University, 1090 Carmack Road, Columbus, Ohio 43210-1002, USA

[7]School of Earth Sciences, 275 Mendenhall Laboratory, The Ohio State University, 125 South Oval 8  Mall, Columbus, OH 43210, USA

*Correspondence to:* Daniela Festi Daniela.Festi@uibk.ac.at

**Abstract.** Dating of ice cores from temperate non-polar glaciers is challenging and often problematic. Yet, a proper timescale is essential for a correct interpretation of the proxies measured in the cores. Here, we introduce a new method developed to obtain a sub-seasonal timescale relying on statistically measured similarities between pollen spectra obtained from core samples, and daily airborne pollen monitoring samples collected in the same area. This approach was developed on a 10 m core retrieved from the temperate firn portion of Alto dell'Ortles glacier (Eastern Italian Alps), for which a five-year annual/seasonal timescale already exists. The aim was to considerably improve this timescale, reaching the highest possible temporal resolution, and testing the efficiency and limits of pollen as a chronological tool. A test of the new timescale was performed by comparing our results to the output (date of layer formation) of the mass balance model EISModel, during the period encompassed by the timescale. The correspondence of the results supports the new sub-seasonal timescale based on pollen analysis. This comparison also allows to draw important conclusions on the post depositional effects of meltwater percolation on the pollen content of the firn core, as well as on the climatic interpretation of the pollen signal. Finally, we provide an example of useful application of the timescale related to the direct comparison of measured meteorological parameters and the stable isotopes composition of the core.

**Keywords:** cryopalynology, ice core, chronology, paleoclimatology

## 1.      Introduction

Ice core dating is crucial for the interpretation of the paleo-climatic and paleo-environmental proxies contained in glacial archives. When combined with the detection of absolute temporal horizons, annual layer counting is the most accurate

technique to date ice cores (Thompson et al., 2013). However, low snow accumulation and/or post-deposition effects (e.g. meltwater percolation) hamper the detection of annual layers, especially in temperate glaciers where annual signals are most often smoothed (Eichler et al., 2001). In addition, no proxies had so far a level of temporal precision that allows studying past seasonal changes in detail, while a sub-seasonal temporal resolution would be desirable to reconstruct these changes in detail.

Several studies conducted on glaciers worldwide have proven that pollen (Nakazawa et al., 2011, 2005, 2004, 2015; Santibañez et al., 2008; Uetake et al., 2006) and stable isotopes (Gabrielli et al., 2008; Haeberli et al., 1983; Thompson, 1980; Dansgaard, 1964; Vareschi, 1934) are valuable proxies to detect seasonality in ice cores. However, annual layer counting from oxygen and hydrogen isotopes ratios is a far more common chronological tool than the palynological approach, the potential of which remains largely unexplored. The reason for the scares use of cryopalynology has functional and conceptual basis. The main

issue is that pollen analyses requires a minimum ice sample size of up to 1 L (e.g., Burogois, 2000), which is problematic to obtain because sample volume from ice cores is very limited, especially when working at high resolution. Yet, studies from the Altai (Nakazawa et al., 2015, 2011, 2005 and 2004) and the European Alps (Festi et al., 2015; Bortenschlager, 1970a and b; Vareschi et al., 1934) suggest that this is still a limit for clean samples obtained from polar ice caps, whereas in low and middle latitude glaciers, the minimum sample size can be reduced to 10-30 mL (Festi et al., 2015; Nakazawa et al., 2011 and

2005 ), thanks to the proximity of the source vegetation and the consequent much greater pollen deposition. An additional limit is that pollen analyses are time-consuming and work-intensive, because they imply manual identification and quantification of pollen grains. In order to overcome this latter issue Nakazawa et al., (2011, 2005 and 2004) adopted a simplified approach, focusing on three main taxa (Pinaceae, Betulaceae and *Artemisia*) that are representative of the three corresponding flowering seasons (spring, early summer, late summer). By doing so, they were able to detect seasonal changes in ice cores from the

Altai (Nakazawa et al., 2015, 2012, 2006 and 2005). Finally, there is little knowledge about the effects of percolating water on the palynological signal and only two studies directly addressed this issue, obtaining contrasting results (Ewing et al., 2014; Nakazawa and Suzuki, 2008).

In Festi et al. (2015) we developed an efficient method to detect seasonality employing a 10 m shallow core extracted in June 2009 from the Alto dell'Ortles glacier (South Tyrol, Italy) (Fig. 1) and we improved the existing timescale based on the isotopic

composition of the core (Gabrielli et al., 2010). In Festi et al. (2015) we conducted accurate taxonomical identification and implemented a statistical approach consisting in performing a principal component analysis (PCA) on pollen concentration values, and extracting the three principal components (PC) indicative of the three flowering seasons. As each PC summarizes the seasonal information of the pollen assemblage, score values of PC indicate seasonal/annual patterns and enable the identification of seasonal and annual firn layers (Fig.2 a). In Festi et al. (2015) we also discussed that the main pollen input on

the glacier likely comes from the near valleys, as the pollen spectra from the ice samples and from valley floor at Solda's monitoring station (Fig. 1) are very similar.

The present study uses the palynological data discussed in Festi et al. (2015) to develop  a new refined and innovative pollen-based method to date ice core samples at a sub-seasonal resolution. The aim was to improve the chronology enhancing resolution, dating efficiency, coherence of seasonal patterns (i.e. overlapping of components).

Combined with mass balance modelling, the new high resolution results also provide new insights into the processes controlling the formation and preservation of the palynological signal in firn and ice. Finally, we give an example on the potential application of the new pollen based timescale by applying it to the isotopic composition of the core, hereby performing a direct comparison between meteorological parameters (i.e. temperature) and isotopes measured in the core, allowing to gain a new insight on the processes of formation of the isotopic signal.

## 2.    Study site

The Alto dell'Ortles is the highest glacier of South Tyrol (Italy) in the Eastern European Alps (46° 30' 32'' N, 10° 32' 41'' E) (Fig. 1), ranging in altitude between 3018 and 3905 m a.s.l. and covering an area of 1.07 km². The maximum glacier thickness is about 75 m (Gabrielli et al., 2012) and encompasses the last ~ 7 kyr (Gabrielli et al., 2016). In its upper part the glacier is polythermal, with temperate firn and cold ice underneath (Gabrielli et al., 2012). The local climate is dry and continental, characterized by a mean annual precipitation of 800-950 mm y$^{-1}$ at the valley floor in Solda (Adler, 2015). This study focuses on the uppermost 10 meters of firn accumulated on the glacier from spring 2005 to June 2009, that have been retrieved using a lightweight hand auger at 3830 m a.s.l.. Our previous research (Festi et al., 2015; Gabrielli et al., 2010), proved that the seasonal/annual signature of pollen and stable isotopes is well preserved in these shallow firn layers. Figure 2(a) shows the formerly published timescale (Festi et al., 2015), which is based on the palynological analyses of 103 continuous ~10 cm samples along the core.

In this study, we also use air temperature data recorded by a standalone data logger placed on the Mt. Ortles at 3835 m a.s.l., as well as meteorological (air temperature and precipitation) and airborne pollen data collected at the meteorological station of Solda (Festi et al., 2015). The station is located 4.5 km northeast of the Mt. Ortles at an altitude of 1850 m a.s.l. (Fig. 1).

## 3.     Methods

This section describes i) the novel pollen-based method "from depth-to-day" developed to obtain a high resolution timescale for the 2009 Mt. Ortles shallow firn core, and ii) the approach used to obtain a core layer dating by using a mass balance model (EISModel), which serves as a comparison for the newly developed palynological timescale.

### 3.1.     High resolution pollen-based timescale: the depth-to-day method

Details on the composition of the Mt. Ortles and Solda's pollen assemblages are reported in Festi et al. (2015). Here, we use these data to develop an enhanced resolved chronology at sub-seasonal timescale. The extensive pollen analyses carried out on the 103 samples obtained from the 10 m core provide a high diversity of 64 pollen types, including the main pollen types characterizing the forest vegetation in the region (e.g., *Pinus* sp., *Picea*, *Fagus*, *Corylus*, *Betula*, *Fraxinus*, *Quercus*), as well as herbs (e.g. Poaceae, Chenopodiaceae, Asteraceae, Cannabaceae). The Ortles pollen assemblages proved to be representative for the regional vegetation, and to be comparable with airborne assemblages recorded at the Solda aerobiological station during the years 2008 to 2010 (Festi et al 2015). Solda's airborne pollen data provided crucial daily information about the timing of local flowering of different plant taxa and of the daily changes in airborne pollen concentration for the years 2008 to 2010. Every plant in the region releases pollen during a certain period of the calendar year, and this is repeated on an annual cycle. However, the onset of flowering may differ by several days (1 to 7 depending on the species; Festi et al., 2015) from year to year, due to different weather conditions.

Solda's airborne pollen samples are characterized by their specific pollen content on a specific day of the year (DOY), while each of the 10 cm sequential Mt. Ortles samples is characterized by its pollen assemblage at a specific depth in the firn core. The three-year palynological dataset of the Solda's aerobiological station has been considered as a representative calibration data-set to define the flowering DOY for in the entire period covered by the 10 m Ortles firn core (2005 to 2009). For every 10 cm sequential Mt. Ortles sample, the three Solda's airborne samples with the highest similarity (one per each year of the three years Solda dataset) were selected, using the Jaccard similarity index (Jaccard, 1901). This index was chosen because it is asymmetrical and hence avoids the double zero problem, i.e. it excludes similarity in case of absence of a pollen type from pairs of compared samples. The Jaccard similarity index was calculated with the SPSS software obtaining a matrix of similarity indices. Indices typically presented values scaled from zero to one: the higher the value, the greater the similarity between two samples. The lower boundary for the Jaccard index was set at 0.5 to ensure high similarity and avoid possible mismatches.

Three potential DOYs were obtained for every Mt. Ortles sample. For each sample, the average DOY and uncertainty (one sigma) have been calculated (Table 1, Fig. 2 (b)). Mt. Ortles samples having pollen concentrations reflecting "winter" season (<0.5 grains ml$^{-1}$,) were excluded from the analyses, as Solda's airborne station observes only pollen distributions during the vegetation period (March - October). Therefore, the added value of the sub-seasonal dating method mostly lies in spring and summer months, i.e. the ablation season on glaciers. This time matching between flowering and ablation is one of the reasons why we implemented this procedure.

In summary, by coupling the Mt. Ortles firn samples with the most statistically similar assemblage of the Solda's airborne samples, we establish a link between pollen deposition at a specific sample depth on Alto dell'Ortles and a specific DOY. This "space-for-time" substitution (depth-to-day) by pollen is the key concept of the new dating technique developed. This method is based on the assumption that there is no time lag between the flowering in Solda and the pollen deposition on the glacier, thanks to the efficient uplift of pollen grains by thermic wind (Barry and Chorley, 2009). Depth values were converted to water equivalent values to enable comparison with EISModel results, using a polynomial function fitted to the 2009 firn core density sampled at 10 cm depth intervals. Samples are ordered (Table1) according to their increasing depths from the top of the core, and increasing w.e. from the bottom of the core.

## 3.2.    EISModel

Here we briefly describe the essential parts of the mass balance model used in this study (Carturan et al., 2012a). The cumulated mass balance from 2005 to 2009 at the coring site was calculated using EISModel (Carturan et al., 2012a; Cazorzi and Dalla Fontana, 1996). Before being included in the model, the raw meteorological data recorded on Mt. Ortles and at Solda, were checked and validated against other meteorological weather stations located in the proximity of Mt. Ortles (Madriccio at 2825 m and Cima Beltovo at 3328 m). The precipitation data recorded at Solda were corrected for gauge undercatch errors, using the procedure described in Carturan et al. (2012b).

The mass balance model simulates accumulation and melt processes at hourly time steps. Snow accumulation was calculated from the precipitation data recorded at Solda, extrapolated to the elevation of the study site using a Precipitation Linear Increase Factor *PLIF* (% km$^{-1}$), to account for the increase of precipitation with altitude. Combining Solda's precipitation with measurements of snow water equivalent performed in the area of Madriccio weather station at the end of winter 2013 and 2014, at an altitude of 2700-2800 m (Carturan L., unpublished data), has enabled the calculation of an average *PLIF* of 50% km$^{-1}$, which was assumed to be valid also for the period 2005-2009. Effects from preferential snow deposition, sumblimation and erosion by wind were not taken into account in extrapolating precipitations at the core site, assuming they compensate each other. In absence of direct observations, this is a reasonable assumption and its possible effects are discussed further on in the paper. Internal accumulation due to refreezing of percolating water was not calculated because it is negligible in temperate-firn layers (March and Trabant, 1997; Cogley at al., 2011; Zemp et al., 2013)

Ablation was calculated by means of an enhanced temperature-index approach, using the clear-sky shortwave radiation computed from a Digital Elevation Model (LiDAR survey of 2005, provided by the Province of Bolzano) as a distributed morpho-energetic index (Carturan et al., 2012a). The melt rate $MLT_t$ (mm h$^{-1}$) was calculated for each hour (t) as follows:

$$MLT_t = RTMF \cdot CSR_t(1 - \alpha_t) \cdot T_t \tag{1}$$

where $T_t$ (°C) is the air temperature, $CSR_t$ (W m$^{-2}$) is the clear sky shortwave radiation and $\alpha_t$ is the surface albedo (calculated in function of $T_t$, Carturan et al., 2012a). *RTMF* is a calibration coefficient called Radiation-Temperature Melt Factor (mm h$^{-1}$°C$^{-1}$W$^{-1}$ m$^2$). The $T_t$ at the study site was calculated from the air temperature measured at Solda, applying a monthly-variable lapse rate ranging from -4.6°C km$^{-1}$ in November to -6.8°C km$^{-1}$ in July (mean value = -5.9°C km$^{-1}$), as calculated between Solda and the air temperature logger placed at the core site at 3835 m on Alto dell'Ortles in the period from October

2011 to May 2014. The RTMF value was obtained by using mass balance measurements (snow depth soundings and density measurements in snow pits) carried out at the core site on June 12 and August 31, 2009.

For each snow layer deposited (i.e. the water equivalent that accumulates at the surface of the snowpack during an hourly time step), the model provides its time and date of formation as well as the air temperature during its deposition.

## 4.   Results

### 4.1.   Pollen based timescale

Results of depth-to-day match of firn and Solda's samples are shown in Table 1 and Fig. 2(b). Each date illustrates the time period encompassed by the sample, with the amplitude of the error bars indicating the number of days potentially included in each sample. The surface sample was dated to the 6[th] of June 2009, in good agreement with the fact that the core was extracted in the first half of June. The dates cluster in 5 groups representing 5 vegetation periods, and are distributed in chronological order within each year. Few inversions are present (i.e., samples at 78 cm w.e. in 2006; 175 cm w.e. in 2007; 324 cm w.e. in 2008; 434, 448 and 452 cm in 2009), but the estimated dates are generally within the uncertainty of adjacent samples. Such inversions are reported in Table 1, as they potentially represent disturbances in the snow deposition (see discussion paragraph 5.2). We observe a substantial difference in the pollen distribution patterns in the snow, both within and among flowering seasons. Flowering seasons stand out distinctively as layers with high pollen concentration, variable inter-annual thickness and seasonality patterns. According to their thickness and vertical distribution of pollen, the flowering years cluster in two groups: 2005 and 2006 vs. 2007, 2008 and 2009. In details, the flowering seasons of 2007 and 2008  correspond to very thick firn layers (76 and 65 cm w.e. respectively), into which pollen is distributed with a clear seasonal pattern. In contrast, the flowering seasons of 2005 and 2006 are characterized by: i) a significantly lower firn thickness (12 and 34 cm w.e. respectively), ii) the occurrence of a thin lower layer with a distinct spring pollen content, and iii) a thin upper layer containing mixed spring/summer pollen. On the contrary, the non-flowering seasons (October to February) are clearly visible as firn layers which are free (or nearly free) of pollen and present significant differences in thickness. Winter 2007/08 is the thinnest, with only 33 cm w.e., followed by 2006/07 with 44 cm w.e., 2005/06 with 60 cm w.e. and finally 2008/09 with 91 cm w.e..

### 4.2.   EISModel ouput

The cumulated mass balance simulated by the model shows high variability of accumulation and melt rates during the period from 2005 to 2009 (Fig. 3). In particular, there is a strong difference between the first two years (2005 and 2006), during which snow accumulation mostly occurred in late summer after weeks with high ablation, and the last three years (from 2007 to 2009), characterized by higher accumulation rates and a much lower summer ablation. In 2005, the ablation removed snow layers accumulated between the 9[th] April and the 31[st] of July. The same happened in 2006 to the snow layers accumulated between the end of March and the end of July.

Given that the precipitation regime in the Mt. Ortles area is characterized by a winter minimum and a summer maximum (Schwarb, 2000), corresponding maxima and minima in the snow accumulation rate are expected at the study site, in absence of significant ablation. This behaviour was indeed modelled in the hydrological years 2006 - 2007 and 2007 - 2008. In 2008 - 2009 the modelled accumulation rate was high also during winter, as a result of abundant winter precipitation in the Italian Alps.

## 5.    Discussion

### 5.1.  Improvements in the pollen timescale

The depth-to-day method and the PCs approach (Festi et al., 2015)provide converging evidences that the core encompasses 5 vegetation periods, that can be assigned to specific years by counting back from the core's extraction date in June 2009 (Fig. 2, panels (a) and (b)). However, the new method presents advantages at different levels: i) it provides a sub-seasonal time resolved dating; ii) it estimates the timespan encompassed by each dated sample (represented by the amplitude of the error); and iii) it resolves the main issue of the PCs method, namely extracting a coherent chronological succession when PCs overlap, e.g. in years 2005 and 2006. In the year 2005 the overlapping of principal components (PCs) makes the interpretation of the seasonality problematic. On the contrary, the depth-to-day method provides three distinctive dates in chronological order, encompassing a time period from April to July (Fig 2(b)). A similar situation is found in the year 2006, which mainly corresponds to an overlapping of PCs between 80 and 85 cm w.e. of depth. This overlapping follows a very low rise in the spring component and precedes an equally low increase in the late-summer component. The depth-to day method provides a better chronological succession of the samples, by dating the first sample of 2006 to March, a second cluster of three samples to May-beginning of June (corresponding to the PCs overlapping) and the final sample to the mid-end of June. The central cluster of dates includes also an inversion, likely indicating disturbances in the pollen and/or in the snow deposition, possibly caused by melting or wind erosion and redeposition etc. (see also section 5.3). Also in 2007 the succession of dates provides a more straightforward timescale than the components do, solving the case of overlapping components between 160-180 cm w.e. of depth. The ability of the depth-to-day method to produce a detailed dating for layers otherwise characterized by the PCs overlapping suggests that this method is more efficient in detecting changes in the pollen assemblage. In general, component overlapping can be expected because the flowering of the different plants occurs in a continuum, without pauses between spring, early-summer and late summer season, and therefore without sharp changes in the airborne pollen assemblage. The only break in the flowering is represented by the autumn and winter, as in this period there is no local pollen production, leading to firn layers with no, or sporadic pollen occurrence. In addition, the PC overlapping may also result from sampling two flowering season in a single snow/firn sample.

In summary, the day-to-depth method significantly improved the timescale by Festi et al. (2015), providing a clear and intuitive representation of the chronological succession, showing a better efficiency in detecting changes in pollen assemblages that allows establishing a sub-seasonal resolution timescale, and providing the estimation of the number of days included in each dated sample.

## 5.2. Comparison of the pollen and modelled timescales

Application of the "depth-to-day" pollen concept and mass balance modelling provides two independent dating of the snow and firn layers of the 10 m firn core drilled in June 2009. Layer dating obtained with EISModel and pollen dating are in very good agreement (Pearson correlation coefficient r=0.99; $p<0.01$) as shown in Fig. 3. The chronological development of snow accumulation as modelled by EISModel matches very well with the absolute dates provided by the pollen-based timescale (Fig. 3). This can be partially due to the compensating effects of a small overestimation of the modelled ablation during summer and the higher accumulation during winter (Fig. 4). The most likely cause of such behaviour is the simplified modelling of snow accumulation, which does not account for redistribution processes. Nevertheless, , the modelling approach looks robust, as revealed by validation tests carried out in the period from August 2008 to August 2013. Without parameter adjustments, the EISModel reliably calculated the mass balance measured in 10 different dates at the core site (+10% cumulated error in 5 years, 23 cm w.e. RMSE for single dates).

The two methods provide converging evidence that two distinct types of warm seasons occurred during the investigated period, i.e. summers with high ablation in 2005 and 2006, and summers with high accumulation in 2007 and 2008 (Fig. 3 and 4). The high agreement between pollen dating and EISModel results further confirms that in the Ortles region pollen grains are very efficiently transported upward from the vegetation source to the glacier with no/negligible time lag. This is due to the vicinity of the vegetation to the ice body and to the steep altitudinal gradient (Fig.1). Such efficiency cannot be assumed to be a common

feature for every glacier-periglacial environment. In fact, Nakazawa et al. (2015) observed a significant delay (about one month) between pollen production and its deposition on a glacier in the Mongolian Altai.

Comparing the detailed chronological distributions obtained by the palynological methods and by the EISModel, is of help in the interpretation of problematic features such as the pollen timescale inversions. Theoretically, chronological inversions can potentially have different origins: statistical and/or physical. They might be a statistical artefacts due to the limited time period (three years) covered by the available data from the aerobiological station of Solda. A longer training period for our palynological method would probably have enabled a better dating accuracy. On other hand, there can also be physical processes that have altered the original deposition of pollen and snow, for example the redistribution of surface snow layers caused by wind erosion and redisposition, and the mixing of pollen in melting layers. The high-altitude location of the Ortles core site favors strong wind redistribution because it is not sheltered by the surrounding relief, and because the low temperature keeps the snow dry (and therefore mobilizable) for long periods, also in spring and summer. The effects from surface melt and water percolation are discussed in the following section.

### 5.3.    Melt water effect on the pollen signal

For the year 2005 and 2006 the EISModel calculations indicate the occurrence of strong summer ablation, leading to the removal of spring and early summer layers for a thickness of about 30 cm w.e. The occurrence of a 0.45 cm thick ice lens at the bottom of the core, and a thick 6.3 cm lens at 50 cm w.e. (Fig 2d) are indications of water percolation and refreezing due to summer melting. Despite water percolation, dating of the 2005 samples provided dates which are in chronological order, with no evidence of pollen displacement. In the 2006 sequence, the only hint for a minor displacement of the pollen is a cluster of three samples dated to May-June, which presents small inversions. While it is hard to assign the origin of such inversions to either percolating water or a limited training set for the depth-to-day method, a minor displacement of the pollen within these layers cannot be completely excluded. On the contrary, a major displacement can be excluded because there is no evidence of pollen displacement downward to the early spring layers in both 2005 and 2006, as well as in the 2005/06 winter strata, despite the fact that ablation reached 30 cm w.e.in summer 2006. In fact, pollen concentration in this winter layer remains below 0.5 pollens mL$^{-1}$. This value is similar to winter pollen concentration values of the hydrological years characterized by lower summer ablation (i.e. 2007 and 2008), and it is in agreement with the low winter pollen accumulation reported from the western Alps (Haeberli et al., 1983). A further indication of water percolation and refreezing comes from a 9 cm ice lens detected at 150 cm w.e., corresponding to the first datable sample of the year 2007. According to EISModel calculations, summer ablation in 2007 was lower than in 2005 and 2006, and there are no clear evidences of downward pollen transportation. In 2007 the dates are, indeed, in chronological order and include inversion of minor entity. Based on these results, it is possible to infer that during periods with high summer ablation, as during the warm seasons 2005 and 2006, pollen grains are not transported by percolation to lower layers, but mainly concentrate into surface layers similarly to dust and debris (Gabrielli et al., 2014).

Our results on the resilience of pollen to percolating water also agree with observations by Nakazawa and Suzuki (2008) in a snowpack study on the Norikura Highlands (1590 m asl) in Japan, where they found that during melt phases pollen concentrates on the surface of the snow and is not transported downward by meltwater. This outcome supports the idea that pollen grains are too big (5-200 µm) to be easily displaced by percolation within snow or firn (Nakazawa et al., 2004). Nakazawa et al (2015) in their study on the Potanin Glacier (Mongolian Altai) postulated that the spring pollen layers could be enriched in pollen as a result of heavy summer melting. Also in this case, no evidence of pollen transport was found in the winter layers. In contrast, Ewing et al. (2014) obtained different results in a laboratory experiment, simulating post-depositional processes with different glacier snowpack conditions using Styrofoam coolers (60 x 30 x 30 cm) filled with natural winter snow accumulation. In this study, they observed a major vertical displacement of pollen grains concluding that meltwater highly affects pollen distribution.

These divergent conclusions on the effect of percolating water on pollen grains point to the fact that more specific studies on this phenomenon are needed, as it is likely influenced by laboratory design conditions, the natural interplay of local micro-climatic conditions, physical characteristics of snow and firn, and possibly pollen grain size and shape. For example, on the Alto dell'Ortles, fresh winter snow usually accumulates in windy conditions and has a density of 300 kg m$^{-3}$ (Gabrielli et al., 2010) and similar values are reported by Nakazawa et al (2012, 2015). In contrast, snow deposited at other sites with low wind conditions can have a lower density (50-70 kg m$^{-3}$) and higher porosity, which would facilitate the vertical dislocation of pollen grains. For the Alto dell'Ortles glacier, a major displacement of pollen grains by meltwater percolation can be ruled out during the studied time period. The few pollen grains retrieved in the cold season layers are likely associated to the occasional input of redeposited regional pollen grains or long-distance transport of pollen brought by windy events, which could carry pollen from the near Mediterranean region, where the flowering season is longer. A local input of redeposited pollen during winter is unlikely at the study site, because during the cold season the local atmospheric boundary layer generally lies below 2000 m a.s.l., trapping pollen and pollutants in the lowermost layers of the troposphere (Gabrieli et al., 2011).

The combined use of the pollen and EISModel timescale further corroborates the finding by Festi et al. (2015), which improved the dating of the 2009 shallow core obtained from chemistry-based seasonality (Gabrielli et al., 2010) that is more likely to be affected by meltwater percolation. Therefore, an approach that combines at least two of these methods turns out to be a more reliable approach for dating firn cores from temperate glaciers in the Alps. This may be valid also for cores retrieved from other ice bodies located in an environmental setting similar to the European Alps, where the vegetation is close to the glaciers and leads to abundant pollen deposition.

### 5.4.     The potential of pollen for qualitative climatic reconstruction

The comparison between pollen content of sequential firn samples and cumulative mass balance modelling (Fig 3 and 4), led us to argue that pollen has a good potential not only for dating, but also for inferring the impact of past climatic conditions in firn and ice cores at seasonal resolution, as already recongnized by Nakazawa et al (2015). Based on the pollen content in the Mt. Ortles strata, three main types of pollen assemblages can be identified and correlated with the corresponding seasonal climatic conditions inferred by EISModel:  i) thin pollen rich layers with fairly clear date order but PCs overlapping (i.e. 2005 and 2006). EISModel indicates that such layers are the result of intense summer ablation thus pointing to warm and dry summer periods; ii) thick layers with significant pollen concentration and well-distinguished succession of dates (i.e. 2007 and 2008). For these layers EISModel shows a spring/summer snow deposition characterized by minimal melting, generated  by abundant precipitation and low temperatures; iii) thick layers with no (or nearly absent) pollen, representing snow deposition during the autumn and winter seasons.

Figure 4 shows the correlation between the inferred summer and winter mass balance according to the pollen dates and EISModel. The pollen based seasonal snow accumulation has been defined for each flowering year (summer in graph) as the depth in w.e. encompassed between the first and last datable sample of each year, while winter (non-flowering season) mass balance has been calculated as the w.e. accumulated between consecutive summer (flowering seasons). For a direct comparison, the same dates for each season were used for EISModel. Figure 4 shows that the thickness of the warm/cold season layers dated by pollen analyses and modelled by EISModel gives a fairly good qualitative indication of the amount of seasonal precipitation: summer 2005 and 2006 stand out for the low accumulation in w.e. with both methods; on the contrary summer 2007 and 2008 are inferred by both methods as high accumulation summers. Similarly, among the cold seasons winter 2008-09 stands out for its higher w.e. thickness correlated to exceptionally high precipitation.

Undoubtedly, this is only the first step towards a qualitative climatic interpretation of the pollen signal and further investigations on longer climate series and pollen sequences from glacier cores are required in order support this evidence. Further studies are also required to detect as many combinations of pollen assemblage types-ice layers as possible, and to further correlate them with the corresponding conditions of formation. In fact, there are several other possible combinations,

e.g. thin layers with high pollen concentration formed during dry (but not particularly hot) summer periods, or thick layers with high pollen content but inconsistent or mixed sequence of seasonal components, deriving from relatively low winter accumulation and possible blending of two or more years. Our results suggest that, combining the classical stable isotope method with palynological analyses does not only enhance the accuracy of the ice core dating, but also offers the potential to provide a calibration set to obtain qualitative paleo-climatic information at seasonal resolution. Clearly, the feasibility of this approach in deeper ice cores depends on the amount of ice available for pollen analyses, the resolution achievable during sampling and the condition of no delay between pollen production and deposition. When applying this qualitative method to deeper ice cores a thinning of the layers by compression must also be taken into account while comparing ice or firn accumulated in different epochs. Long term studies on the phenology and historic series of pollen monitoring encompassing the last 35 years show that the shift of the onset/end of the blooming season due to temperature trends are on average of 10 days in Europe (Menzel & Fabian 1999, Nature) as well as in the Ortles region (Bortenschalger & Borthenschlager 2007, Grana). We therefore suggest that this difference is not particularly relevant when dating a deep core, as this value is surely smaller than the number of days encompassed in one deep core sample.

## 5.5.     Application of the pollen based timescale

The application of the new detailed sub-seasonal timescale to other climatic proxies contained in ice cores (e.g. stable isotopes, major ions, dust, etc.) enables a direct comparison with meteorological variables and opens novel possibilities for a better understanding of processes related to the deposition and preservation of specific proxies. An example of a possible application is provided in Fig. 5(a), where the timescale has been applied to the $\delta D$ values obtained from the same shallow firn core retrieved on Alto dell'Ortles. For this purpose, a complete timescale was calculated for the core, in order to date all the snow and firn samples and to plot their isotopic composition as a function of time. For samples belonging to the flowering seasons, absolute dates obtained with the depth-to-day pollen method were used (Table 1 and Fig. 2) and inversions were also included in the timescale. For the cold season layers, in absence of direct measurements of snow accumulation, a simple linear regression of depth vs. time was calculated. By applying this timescale to the $\delta D$ values, a direct comparison between isotopes and mean daily temperature data as recorded in Solda became possible. This is supposed to be meaningful as there is a very high correlation (r = 0.95) between daily air temperature measured at Solda and on the Mt. Ortles shallow core drilling site, during the period from 2011-2014. Figure 5 shows a good correlation (Pearson; r = 0.642; $p < 0.001$) between the mean daily temperature and the measured isotopic composition, which is however not-stationary as it is affected by the smoothing of $\delta D$ in the deepest firn layers (Gabrielli et al., 2010) because of meltwater percolation from the upper layers and partial melting of the snow accumulated during the 2005 and 2006 warm seasons. In contrast, the years 2007 and 2008 show a distinct $\delta D$ summer peak, likely linked to the above-mentioned higher accumulation and lower ablation during those summers.
In conclusion, this is an example that clearly highlights the utility of combining different approaches for investigating the involved physical processes and for interpreting the proxy data contained in paleo-climatic archives.

## 6.     Conclusions

In this study, we have proposed a sub-seasonal timescale for the 10 m Mt. Ortles firn core. The day-to-depth method significantly improved the timescale by Festi et al. (2015), providing a clear and intuitive representation of the chronological succession, proving a better efficiency in detecting changes in pollen assemblages that allows establishing a sub-seasonal resolution timescale, and providing the estimation of the number of days included in each dated sample. The method can be applied to all types of glaciers, regardless of their thermic state, provided the proximity of the pollen source, the existence of flowering seasons (typical of mid-latitudes) or a clear contrast between a flowering and non-flowering season (typical of the tropics and equator due to the alternation of dry and humid season), and the support of modern pollen monitoring data. We

also show that a three- years training set of pollen monitoring is sufficient to provide meaningful comparison with glacier samples. This consideration becomes particularly relevant when working in glaciated regions that are not covered by the pollen allergy network, which is the main source of modern pollen data, e.g. World Allergy Organisation-WAO (worldallergy.org), European Aeroallergen Network-EAN (ean.pollenifo.eu), etc. In this case, a three years monitoring can be included in the cryopalynological study and can take place also after the coring. In alternative, data from the closest stations in the region can be used and a correction for the delay between pollen production and deposition can be applied to the dating model. The method is particularly relevant at the moment because of the increasing interest in ice cores from non-polar areas, as testified by recent launch of the "Protecting ice memory" project, which aims at creating the first archive of glacial ice for future generations. In fact, our approach can be applied to date deeper ice cores, if sampling resolution is sufficiently high. In such cores our method can make a significant contribution to the chronological model also by detecting the transition from annual to non-annual layering in the deeper ice.

The combined use of a mass balance model and pollen-based dating methodology brings compelling evidence that on the Alto dell'Ortles glacier pollen grains are resilient to downward transport by percolating meltwater, also in the case of strong melting as in 2005 and 2006. The independent dating of firn layers by mass balance modelling and pollen match well, and highlight detectable intra-seasonal and inter-annual differences of high altitude snow accumulation rates on Mt. Ortles. More specifically, we found evidence of peculiar types of pollen distribution in firn layers, that may be related to well defined weather types (e.g. warm-dry, warm-humid or cold-humid weather). These results reveal the good potential of pollen for inferring past climatic conditions at a sub-seasonal resolution in ice core records. Finally, we show that a sub-seasonal timescale is valuable because it can be used to investigate the relationship between climatic proxies contained in ice cores and measured meteorological variables for sites where this relationship has not been studied on site. Future studies could focus on joined fieldwork observations of meteorological variables (temperature, snow, wind, etc.) with measurements of snow accumulation/ablation, pollen accumulation, signature of isotopes and other chemical species as well as on monitoring of post depositional effects (snow redistribution, water percolation, sublimation) on the proxies in order to directly test the assumptions made in the present study. This type of approach opens the possibility of gaining new insights into the physical processes involved in the formation and preservation of the signal stored in this paleo-climatic archive.

**Author contributions**

D. Festi, W. Kofler and K. Oeggl performed palynological analyses on the Ortles samples and developed the pollen based timescale. E. Bucher performed the pollen analyses at the aerobiological station of Solda. L. Carturan, F. Cazorzi and F. de Blasi processed the raw meteorological data and mass balance measurements, and carried out the mass balance simulations using EISModel. P. Gabrielli planned the logistic of the field campaign, drilled the firn core and coordinated the processing of the samples. D. Festi, L. Carturan, P. Gabrielli and K. Oeggl prepared the manuscript with contributions from all co-authors.

**Acknowledgements**

We thank the Autonome Provinz Bozen – Südtirol, Abteilung Bildungsförderung, Universität und Forschung for financial support to the project PAMOGIS (Pollen Analyses of the Mt.Ortles Ice Samples). This work is a contribution to the "Ortles project"- a program supported by NSF Awards #1060115 and #1461422 to The Ohio State University, the Fire protection and civil division of the Autonomous Province of Bolzano in collaboration with the Forest division of the Autonomous Province of Bolzano and the National Park of Stelvio. This is the Ortles project publication no. 8. We acknowledge the collaboration in the various phases of the 2009 and following years field operations of Hanspeter Staffler, Michela Munari and Roberto Dinale (Fire Protection and Civil Division of the Autonomous Province of Bolzano) Ludwig Noessig, Elmar Wolfsgruber and Claudio Carraro (Ufficio geologia e prove materiali of the Autonomous Province of Bolzano), Marc Zebisch and Philipp Rastner (EURAC), Jacopo Gabrieli (IDPA CNR-Venice), Roberto Seppi (University of Pavia), Thomas Zanoner (University of

Padova), Karl Krainer (University of Innsbruck), Paul Vallelonga (University of Copenhagen), Michele Lanzinger, Matteo Cattadori and Roberto Filippi (Museo Tridentino di Scienze Naturali) and Silvia Forti (Istituto di Cultura le Marcelline). We thank Michele Bertò for assistance in designing Fig 2. For the logistic support, we are grateful to Toni Stocker (Alpine guides of Solda) and the helicopter company Airway. This is also BPCRC contribution no. 1562.

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

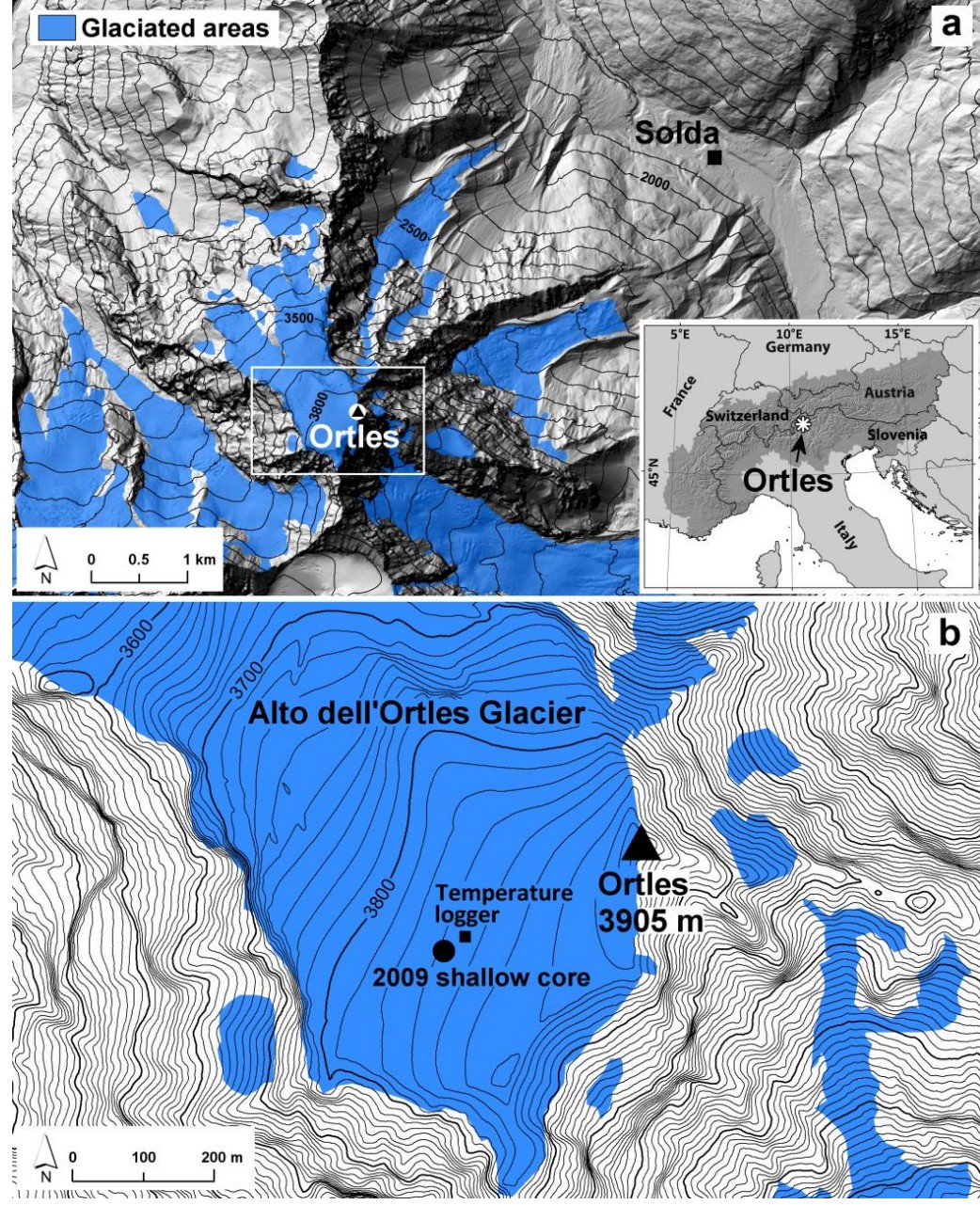

**Figure 1 Geographical setting of Mt. Ortles and of the meteorological station of Solda. b) Close up of the Alto dell'Ortles Glacier, where the 2009 shallow firn core was retrieved (figure adapted from Festi et al., 2015).**

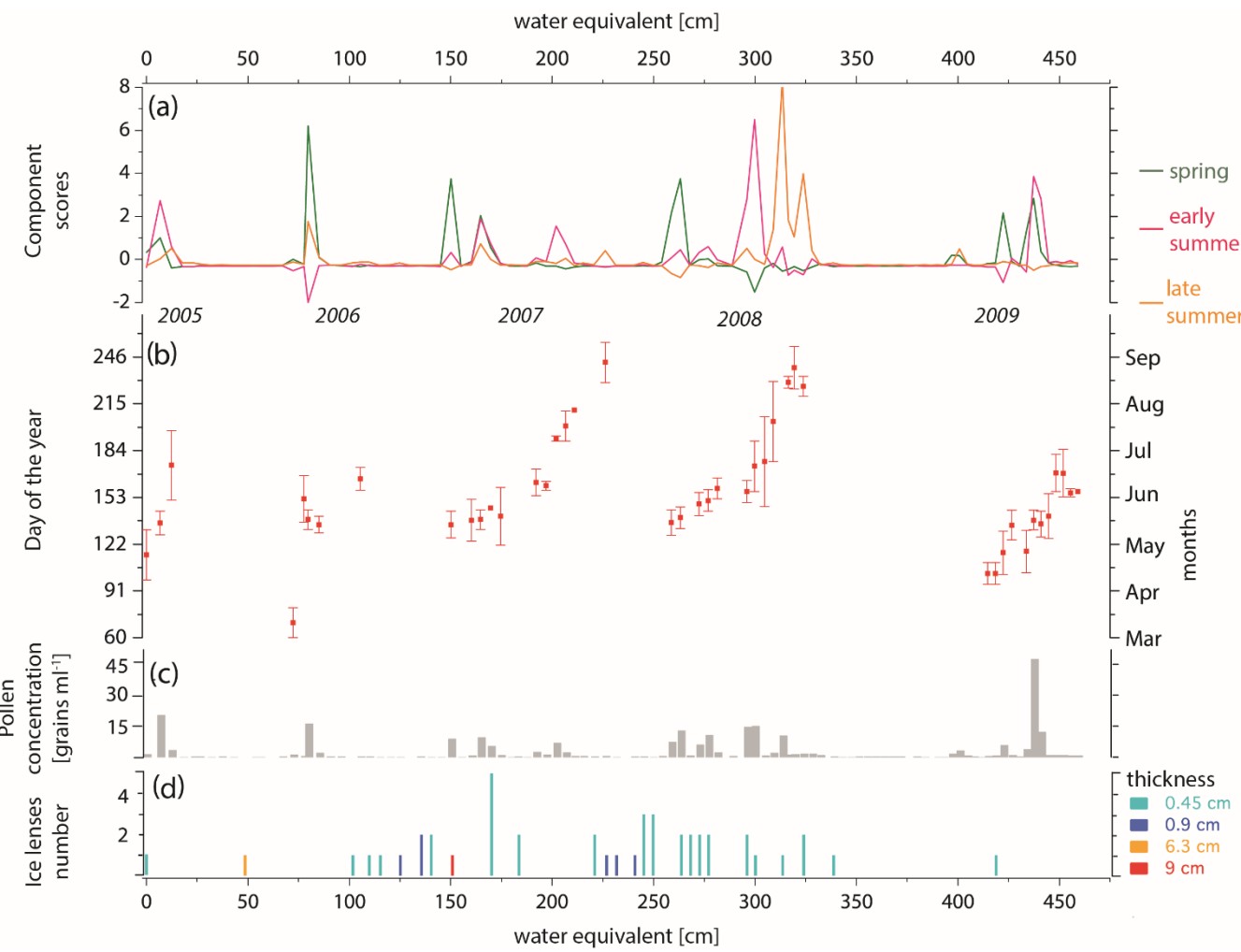

**Figure 2 (a) Mt. Ortles firn core 2009 timescale (Festi et al., 2015) based on Principal Component Analyses. As every principal component (PC) condenses the seasonal information of the pollen spectrum, scores values should to be interpreted as follows: a sample (w.e. depth down) presenting high component scores values for a specific PC is characterized by a pollen content reflecting predominantly the season corresponding to that particular PC. (b) Mean date (day of the year ± 1 sigma) obtained by using the new method developed in this paper. (c) Pollen concentration in the Mt. Ortles 2009 firn core. (d) Number and thickness (colour scale) of ice lenses.**

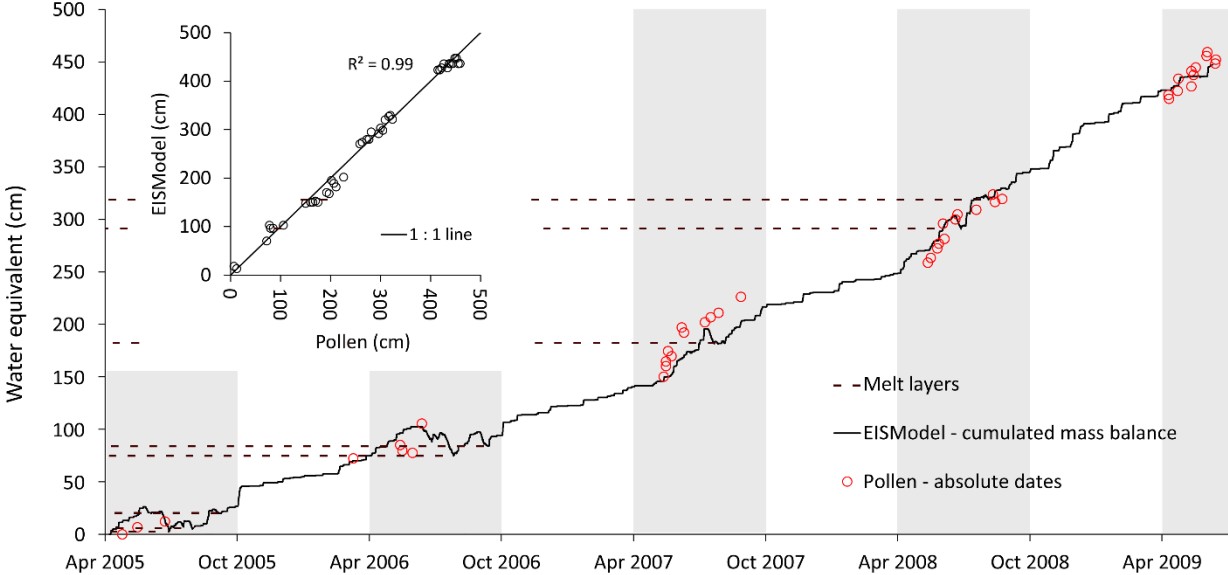

**Figure 3 Comparison between cumulated mass balance modelled by EISModel and obtained by pollen dating. Alternating grey/white bars have a six months duration and roughly indicate the warm/cold season. Horizontal dash lines indicate melt layers.**

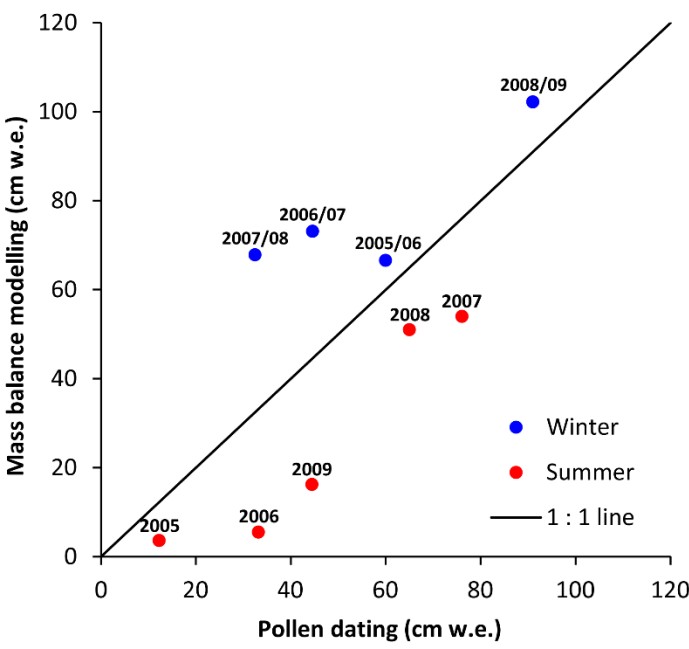

**Figure 4 Comparison of the winter and summer mass balance as obtained by pollen dating and modelling (EISModel).**

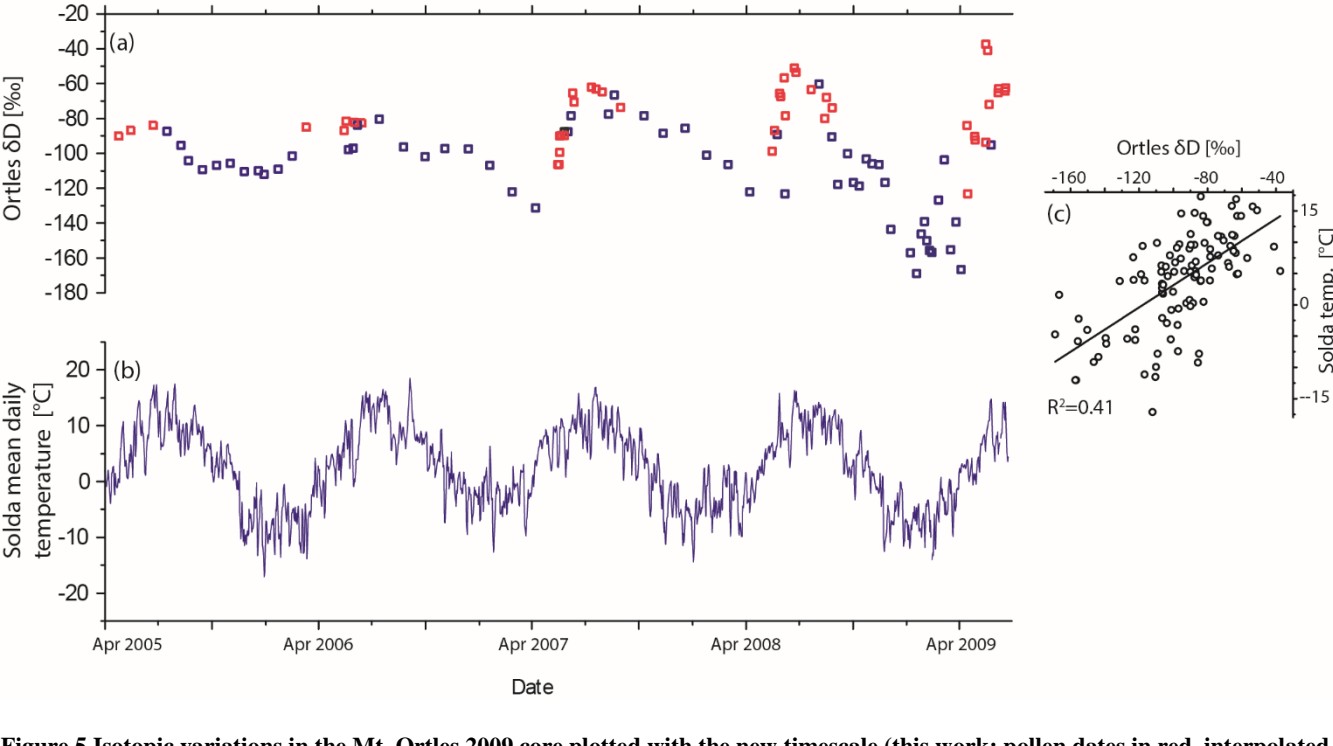

**Figure 5** Isotopic variations in the Mt. Ortles 2009 core plotted with the new timescale (this work: pollen dates in red, interpolated dates in blue) (a) compared with the air temperature measured at Solda during the same period (b); Correlation diagram of Ortles isotopic composition (δD) and Solda's mean daily temperature.

**Table 1. Results of the depth to date matching obtained by similarity analyses between Mt. Ortles snow samples and pollen monitoring data from Solda from the years 2008 to 2010. DOY= day of the year. Samples are ordered according to their depth in the core.**

| Depth (cm) | Water Equivalent (cm) | 2008 (DOY) | 2009 (DOY) | 2010 (DOY) | Mean (DOY) | Date (dd.mm.yy) | SD (±days) |
|---|---|---|---|---|---|---|---|
| 10 | 459 | 157 | 157 | - | 157.0 | 06.06.2009 | 0 |
| 20 | 456 | 159 | 155 | 154 | 156.0 | 05.06.2009 | 3 |
| 30 | 452 | 163 | 157 | 187 | 169.0 | 18.06.2009 | 16 |
| 40 | 448 | 166 | 159 | 183 | 169.3 | 17.06.2009 | 12 |
| 50 | 445 | - | 130 | 151 | 140.5 | 21.05.2009 | 15 |
| 60 | 441 | 132 | 129 | 145 | 135.3 | 15.05.2009 | 9 |
| 70 | 437 | 132 | 137 | 145 | 138.0 | 18.05.2009 | 7 |
| 79.5 | 434 | 132 | 104 | 116 | 117.3 | 27.04.2009 | 14 |
| 99 | 427 | 129 | 129 | 146 | 134.7 | 15.05.2009 | 12 |
| 110.5 | 422 | 132 | 104 | 113 | 116.3 | 26.04.2009 | 12 |
| 120.5 | 418 | 111 | 99 | 98 | 102.7 | 13.04.2009 | 10 |
| 130.5 | 415 | 111 | 99 | 98 | 102.7 | 14.04.2009 | 14 |

| | | | | | | | |
|---|---|---|---|---|---|---|---|
| 355.5 | 324 | 219 | 231 | 230 | 226.7 | 13.08.2008 | 7 |
| 365.5 | 319 | 229 | - | 249 | 239.0 | 26.08.2008 | 14 |
| 372.5 | 316 | 225 | 233 | 230 | 229.3 | 16.08.2008 | 4 |
| 389.5 | 309 | 178 | 231 | 201 | 203.3 | 21.07.2008 | 27 |
| 399.5 | 305 | 211 | 157 | 162 | 176.7 | 25.06.2008 | 30 |
| 410.5 | 300 | 193 | 166 | 162 | 173.7 | 22.06.2008 | 17 |
| 419.5 | 296 | 149 | 159 | 163 | 157.0 | 05.06.2008 | 7 |
| 452.5 | 281 | 152 | 159 | 166 | 159.0 | 07.06.2008 | 7 |
| 462.5 | 277 | 149 | 159 | 145 | 151.0 | 30.05.2008 | 7 |
| 472.5 | 272 | 147 | 142 | 157 | 148.7 | 28.05.2008 | 8 |
| 492.5 | 263 | 132 | 141 | 146 | 139.7 | 19.05.2008 | 7 |
| 502.5 | 259 | 132 | 131 | 146 | 136.3 | 15.05.2008 | 8 |
| | | | | | | | |
| 572.5 | 226 | 233 | 252 | - | 242.5 | 30.08.2007 | 13 |
| 604.5 | 211 | 212 | - | - | 211.0 | 30.07.2007 | 0 |
| 613.5 | 207 | 211 | 191 | 199 | 200.3 | 19.07.2007 | 10 |
| 623.5 | 202 | 194 | 191 | 191 | 192.0 | 11.07.2007 | 2 |
| 633.5 | 197 | 164 | 159 | 159 | 160.7 | 09.06.2007 | 3 |
| 643.5 | 192 | 173 | 157 | 159 | 163.0 | 12.06.2007 | 9 |
| 679.5 | 175 | - | 127 | 154 | 140.5 | 21.05.2007 | 19 |
| 689.5 | 170 | - | - | 146 | 146.0 | 26.05.2007 | 0 |
| 699.5 | 165 | 132 | 138 | 145 | 138.3 | 18.05.2007 | 7 |
| 708.5 | 160 | 129 | 131 | 154 | 138.0 | 18.05.2007 | 14 |
| 728.5 | 150 | 132 | 128 | 145 | 135.0 | 15.05.2007 | 9 |
| | | | | | | | |
| 815.5 | 106 | 167 | 157 | 172 | 165 | 14.06.2006 | 8 |
| 854.5 | 85 | 134 | 141 | 130 | 135.0 | 15.05.2006 | 6 |
| 864.5 | 80 | 132 | 138 | 145 | 138.3 | 18.05.2006 | 7 |
| 868.5 | 78 | 148 | 139 | 169 | 152.0 | 01.06.2006 | 15 |
| 878.5 | 72 | - | 77 | 63 | 70.0 | 11.03.2006 | 10 |
| | | | | | | | |
| 988.5 | 12 | 201 | 162 | 160 | 174.3 | 23.06.2005 | 23 |
| 998.5 | 7 | 132 | 131 | 145 | 136.0 | 16.05.2005 | 8 |
| 1010.5 | 0 | 134 | 104 | 107 | 115.0 | 25.04.2005 | 17 |