# Peer review of "Linking pollen deposition and snow accumulation on the Alto dell'Ortles glacier (South Tyrol, Italy) for sub-seasonal dating of a firn temperate core"

_The Cryosphere, 2016_

## Referee Comment (RC1) · Anonymous Referee #1 · 2 Jan 2017

With the manuscript, the authors try to improve their dating method for ice cores proposed in their previous study (Festi et al., 2015, Journal of Glaciology) to achieve core analyses at sub-seasonal time resolution. They also attempted to argue the accuracy of the new dating method in comparison with time change of surface level calculated at the drilling site by a mass balance model. In addition, they tried to interpret the profile of $\delta$D values of the core based on their detailed chronology established.

The challenge for high resolution analysis is highly evaluated. However, due to a lack of in-situ observation data, it is difficult to judge the argument in this manuscript. Also,

it seems like that the argument is based only on good results obtained by statistical analysis using the SPSS software. The authors should consider more what the data mean and what statistical techniques mean.

The novelty alone cannot warrant publication of this manuscript. Therefore, I recommend the manuscript not to be published.

Detailed comments:

Section 3.1. High resolution pollen-based timescale: the depth-to-day method

P3, L11: Is the sampling interval of 10 cm appropriate for the sub-seasonal time resolution? The authors need to show its grounds.

P3, L25: The authors mentioned that the onset of flowering may differ by several days. How about the peak season and the end of the season? I think those factors also influence the daily changes of airborne pollen concentration and assemblage.

P3, L32: How long does each 10 cm sample accumulate (accumulation time frame)? I wonder if the authors can compare similarity between daily data from Solda and core samples because of the different time scales.

P4, L6: Does the transportation of airborne pollen depend on the species? I wonder if the pollen composition may be kept until pollen deposition on the glacier.

Section 4.1. Pollen based timescale

The authors need to explain more what each date indicates. I wonder if pollen deposition and snow fall on the glacier do not necessarily occur at the same time. Once melting occurs, how do the authors think about the date of snow and pollen in the core?

For example, significant melting occurred in the summers of 2005 and 2006 based on the model calculation. Then, the surface level of snow was reduced to the level on April 9 and the end of March; respectively, as mentioned on P6 L15-16. Therefore, it should be natural to think the ice core lost those parts when there is no internal accumulation

due to refreezing of percolating water as mentioned by the authors on P4 L25 and P6 L14-16.

On the other hand, pollen grains should be gathered on the boundary of the removed layer. Therefore, the pollen concentration and composition in the layer are disturbed from the original state. Actually, a thin layer containing mixed spring/summer pollen is observed in the core as mentioned on P6 L8. After all, I wonder if deciding the date at detailed level does not make sense with such melting core.

Showing stratigraphy of the core should be helpful for readers' better understanding.

Section 5.1. Comparison of the pollen and modelled timescales

The authors need to discuss the accuracy of layer dating obtained with the EISModel by using observed data, for example, the stake observation or automatic snow depth measurement, event signals of dust storm and volcanic eruption, and etc. Otherwise, the authors cannot insist on the legitimacy of the accuracy of the pollen dating.

The point of argument in the following chapters is unclear. The authors need to revise. Instead of those chapters, the authors should devote pages to the discussion for the concept of dating of snow layers and pollen grains after the post-depositional process, and the accuracy of the pollen dating. The suitable sample thickness for such high-resolution time scale should also be discussed. The sampling intervals of 10 cm in this study may be too thick.

Section 5.2. Melt water effect on the pollen signal

The authors need to clarify more the point of argument in this section. As I have mentioned, I think melting affects the position, concentration and composition of pollen grains and the loss of snow layer. Those post-depositional process should lead to disturb dating of layers in an ice core.

P7 L8: Cite original papers. Those were already mentioned in other papers before the study by Gabrielli et al. (2014).

Section 5.3. The potential of pollen for qualitative climatic reconstruction

I wonder if the authors can be more specific in discussing the analysis results by displaying the data obtained because only abstract conception was mentioned here. The studies in Nakazawa and Fujita (2006, Annals of Glaciology) and Nakazawa et al. (2015, Environmental Earth Sciences) may be useful for the discussion in this chapter.

Section 5.4. Application of the pollen based timescale

The authors need to clarify more the point of argument in this section. As the authors noticed, a good correlation between the mean daily temperature and the measured isotopic composition arises from preservation of seasonal variation of the $\delta$D values. Therefore, the effect of re-evaporation or the stable isotope amount effect seems to be small. However, to reconstruct past temperature, the authors need to analyze the data while considering the smoothing of $\delta$D values.

Section 6. Conclusions

P9 L11: The timing of local flowering of different plant taxa and of the daily changes in airborne pollen concentration should be changed under climate change. How do the authors overcome this problem without airborne pollen data when applying this method to date deeper ice cores?

Table 1

The dates in 2005 and 2006 are manifestly inconsistent with the EISModel calculation and the authors' arguments. It needs to be explain more.
* * *

---

## Referee Comment (RC2) · Anonymous Referee #2 · 4 Jan 2017

Dating of ice cores are challenging subject especially in non polar ice caps, where melting can influence the signal. Here the authors aim at using pollen to date a shallow firn core from the South tyrol alps with a day to day resolution by comparing results from ice cores with nearby station data.

It is an interesting approach to use pollen to date ice cores. In a previous paper (Festi, 2015) the authors used the same core and the same pollen data to date the record using PC and PCA methods to compare with airborne pollen samples from Solda. In this paper they use Jaccard similarities with the same airborne samples from Solda.

They also in this paper use their highly resolved record to derive accumulation rates and compare those with a mass balance model, which as input use meteorological station data and move on to judge whether melt layers influence the pollen record.

The main argument is based purely on the statistics, and the authors should consider how the sample depth may relate to the order of the samples. There is quite some indication of inverse orders, which to my opinion is not very well justified.

However if accepting this kind of uncertainty in the dating of the samples, the pollen only arrive in spring/summer making the date of year dating only possible in spring/summer. This is not very clearly stated either.

To me it is unclear that this paper brings much novelty to the already published paper by Festi (2015) in Journal of Glaciology, however I would suggest the authors to make largely rewrite the manuscript and focus more on the novel aspects (as comapred to the Festi (2015) publication, eg. the comparison to the accumulation model and the water isotopes as well as to emphasize the uncertainties and justify the inversions of the dating better.

Specific comments:

Section 3.1 Define the time of year in which you can make depth to day comparisons based on pollen.

Section 4.1, line 6 expand the explanation about inversions. And consider also in Table 1 to explain this inversions, as the table otherwise is very confusing with dates going back and forth.

Section 4.1 in general should be more concise, it is very long and mainly lists the information from table 1

Section 4.1 eg. line 24, 31, 36 with more. The samples are 10 cm thick (?), how can layers be given in precision of cm, eg. 87 cm, 91 cm etc. These numbers should have uncertainties based on the sample sizes.

Section 5.1 line 35/ Figure 3 (correlation figure). I am confused as it looks like you have more data in the correlation part of Figure 3 (eg. points during winter and autumn), where no points exists in the main part of figure 3 for pollen, where do these additional data stem from?

Section 5.2 Discusses melt layers, however nowhere in any figures are the position of melt layers shown. Please add melt layers to relevant figures, eg. as vertical bars in Fig 2 (and Fig 3).

Section 5.4 (figure 5) . Here you discuss the comparison to water isotopes. In the years 2007, 2008 and 2009 the water isotope data from april to July gets very steep, followed by a somewhat not steep slope down to the next winter. This seems like an effect of you having the pollen data very specifically dated in exactly those months. You suggest that those years are fine, but rather 2006 and 2005 are influenced by meltwater percolation from summer into the winter affecting the summer peak. This may very well be, but it does not explain the lack of similarities for the later years, where the steep slope to me looks very artificial. I would suggest you add sme comments about the uncertainty of the dating, it could eg. be explained by later blooming of the pollen shifting the summer bloom and extending the dD peaks to have a more sinusoidal behavior as also observed in the Solda record of temperature.

---

## Editor Comment (EC1) · J.-L. Tison (Editor) · 14 Feb 2017

Thanks to the authors for replying in details to the reviewer comments.

I would like to underline that one of the reviewer recommended not to publish the paper and that the other recommended an "in-depth" rewriting, e.g. focusing more on the new inputs of the paper compared to Festi et al. (2015) and explaining more specifically the inversions in the dating.

A fair numbers of comments of the reviewers were along the lines of my own preliminary comments as an editor, less expert in the matter. I therefore think that it will be

quite challenging to produce a revised manuscript that satisfies the reviewers. If that, however, is the decision taken by the authors, they should know that I will have to re-submit their manuscript to the same reviewers, with a non-negligible chance that they might not be satisfied, given the nature of the comments made and that I might have to finally reject the paper.

I therefore leave the final decision to the authors to submit a revised manuscript in that context.

Best Regards,

Jean-Louis Tison

---

## Author Comment (AC1) · 14 Feb 2017

Authors: We thank the reviewer for his comments and suggestions.

Referee #1: With the manuscript, the authors try to improve their dating method for ice cores in their previous study (Festi et al., 2015, Journal of Glaciology) to achieve core analyses at sub-seasonal time resolution. They also attempted to argue the accuracy of the new dating method in comparison with time change of surface level calculated at the drilling site by a mass balance model. In addition, they tried to interpret the profile of D values of the core based on their detailed chronology established. The

challenge for high resolution analysis is highly evaluated. However, due to a lack of in-situ observation data, it is difficult to judge the argument in this manuscript.

Authors: It is unclear which in-situ observation data the reviewer refers to. In this study, we actually make use of a combined pollen, meteorological and dD data dataset that is unique at such altitude in the European Alps. Daily monitoring of pollen deposition over glaciers would be highly desirable but it is unfeasible for logistic and economic reasons.

Referee #1: Also, it seems like that the argument is based only on good results obtained by statistical analysis using the SPSS software. The authors should consider more what the data mean and what statistical techniques mean.

Authors: The argument is based on solid and widely used statistical methods, whose use is mandatory considering the large amount of in-situ observation data used in this study. On the other hand, converging independent evidence from palynological methods and from glacier mass balance modelling is extremely unlikely a mere artefact or coincidence (e.g. correlation of pollen and Eismodel results has a correlation coefficient r2=0.99). We would appreciate further and more specific indications by the reviewer, to improve our paper.

Referee #1: The novelty alone cannot warrant publication of this manuscript. Therefore, I recommend the manuscript not to be published.

Authors: We respectfully argue that the method and the interdisciplinary approach are unpreceded and show a very high potential in combining pollen with mass balance models. We believe this paper makes an important contribution to ice core science by i) providing a new high-resolution dating tool, ii) creating a bridge between biology and physical ice core science, iii) process understanding, and iii) inspiring researchers to replicate and develop new methods. We have now emphasized these aspects also within the text.
Referee #1:Section 3.1. High resolution pollen-based timescale: the depth-to-day method P3, L11: Is the sampling interval of 10 cm appropriate for the sub-seasonal time resolution? The authors need to show its grounds.

Authors: The sampling interval is appropriate as demonstrated by the fact that it allowed a sub-seasonal time resolution. This was already illustrated in previously published papers (Gabrielli et al 2010, Festi et al 2015, Kirchgeorg 2016)

Referee #1: P3, L25: The authors mentioned that the onset of flowering may differ by several days. How about the peak season and the end of the season? I think those factors also influence the daily changes of airborne pollen concentration and assemblage.

Authors: The peak in the airborne concentration of a pollen type in the air during the flowering season (above called "peak season") might also differ by a few days each year. However, this is irrelevant for the method because we compare only the presence of the airborne concentration of pollen types. The end of the season also affects the daily assemblage. These differences are taken into account by comparing the ice assemblages with all the monitoring years available and by adding the uncertainty to the date determined.

Referee #1: P3, L32: How long does each 10 cm sample accumulate (accumulation time frame)? I wonder if the authors can compare similarity between daily data from Solda and core samples because of the different time scales.

Authors: There is no homogeneous snow accumulation on Alpine glaciers, as precipitation regimes are not constant throughout the year. The point of the paper is indeed to determine the time period encompassed between different samples, which is virtually always different. Changes to the manuscript: We now specify in section 4.1 that the uncertainty gives an indication of the number of days encompassed by a sample.

Referee #1: P4, L6: Does the transportation of airborne pollen depend on the species?

I wonder if the pollen composition may be kept until pollen deposition on the glacier.

Authors: Given the high correspondence of species found in the ice and at the pollen monitoring station we assume (as already presented within the manuscript and in Festi et al 2015) that the upward transport of all relevant pollen types is equally efficient and does not fractionate the assemblage. To reduce this potential bias we choose the closest and highest pollen monitoring station in the region. For logistic and financial reasons it is not possible to establish a daily pollen monitoring on the Ortles glacier itself. Typically an automatic pollen trap used for daily pollen monitoring requires weekly maintenance (i.e. collect the weekly cylinder on which the pollen is trapped, placing the new cylinder, mechanically recharging the device). Authors, Changes is the manuscript: We added the citation to support species correspondence in section 3.1.

Referee #1: Section 4.1. Pollen based timescale The authors need to explain more what each date indicates. I wonder if pollen deposition and snow fall on the glacier do not necessarily occur at the same time. Once melting occurs, how do the authors think about the date of snow and pollen in the core? For example, significant melting occurred in the summers of 2005 and 2006 based on the model calculation. Then, the surface level of snow was reduced to the level on April 9 and the end of March; respectively, as mentioned on P6 L15-16. Therefore, it should be natural to think the ice core lost those parts when there is no internal accumulation due to refreezing of percolating water as mentioned by the authors on P4 L25 and P6 L14-16. On the other hand, pollen grains should be gathered on the boundary of the removed layer. Therefore, the pollen concentration and composition in the layer are disturbed from the original state.

Authors: Pollen deposition occurs also between snow fall events. This is not a problem for the method as we aim to detect the time period encompassed by the single sample and not to reproduce the date of the single snow event. Once that some melting occurs pollen grains typically remain on the top of the surface layers as also shown by Nakazawa and Suzuki 2008. In years characterized by summer melting, as 2005 and

2006, the dating pattern points to a hot and dry summer season, while the spring layer remains intact. In this way, even when melting occurs the method provides qualitative climatic information.

Referee #1: Section 4.1. Pollen based timescale. Actually, a thin layer containing mixed spring/summer pollen is observed in the core as mentioned on P6 L8. After all, I wonder if deciding the date at detailed level does not make sense with such melting core.

Authors: The critical point of the component method is that the flowering of the species occurs in continuum. However, in order to detect the seasonality in the snow layers we need to create discrete groups of pollen representing seasons. The components of those groups overlap because, for example, spring species do not all start flowering the same day or finish the same day. The same is valid for early and late summer taxa. Furthermore, there is no "flowering pause" between the seasons (except winter). We prove that the new method is more accurate as it is capable of detecting finer changes in the assemblage in comparison to the previews PCs method. This is a significant improvement on the chronology and allows to derive climatic information.

Referee #1: Showing stratigraphy of the core should be helpful for readers' better understanding.

Changes in the manuscript: We added stratigraphy in the Fig 2 with special regard to the ice lenses.

Section 5.1. Comparison of the pollen and modelled timescales Referee #1: The authors need to discuss the accuracy of layer dating obtained with the EISModel by using observed data, for example, the stake observation or automatic snow depth measurement, event signals of dust storm and volcanic eruption, and etc. Otherwise, the authors cannot insist on the legitimacy of the accuracy of the pollen dating.

Authors: The accuracy of EISModel calculations has now been assessed comparing

the calculations with mass balance observations carried out in the period from 2009 to 2013 at the study site. Changes in the manuscript: In session 4.2 we added a statement about the accuracy of Eismodel.

Referee #1 The point of argument in the following chapters is unclear. The authors need to revise. Instead of those chapters, the authors should devote pages to the discussion for the concept of dating of snow layers and pollen grains after the post-depositional process, and the accuracy of the pollen dating. The suitable sample thickness for such high resolution time scale should also be discussed. The sampling intervals of 10 cm in this study may be too thick.

Authors: We now expanded the discussion addressing possible post-depositional effects. The sampling resolution is sufficient since we were able to obtain a sub-seasonal timescale and this has been pointed out. Further evidences supporting the fact that the sampling resolution is adequate are exemplified in previously published papers (Gabrielli et al 2010, Festi et al 2015, Kirchgeorg 2016). Changes in the manuscript: In session 5.1 we expanded the discussion section addressing post depositional effects and dating precision in session as suggested.

Referee #1: Section 5.2. Melt water effect on the pollen signal. The authors need to clarify more the point of argument in this section. As I have mentioned, I think melting affects the position, concentration and composition of pollen grains and the loss of snow layer. Those post-depositional process should lead to disturb dating of layers in an ice core.

Authors: In this paragraph we now provide the evidence that pollen is not easily transported downwards by melting water in the studied core. For example, it is striking that we do not observe a transport of pollen grains through winter layers (as determined also by stable isotopes) even in years where summer melting did occurr. In fact winter layers are markedly depleted in pollen grains providing strong support of the substantial immobility of pollen grains when meltwater percolation occurs. Changes in the

manuscript: We now better discuss the issue of the possible pollen mixing in 2006 in section 5.1.

Referee #1: P7 L8: Cite original papers. Those were already mentioned in other papers before the study by Gabrielli et al. (2014).

Authors: Gabrielli et al 2014 is an adequate citation as it reports direct and recent observation of the phenomenon. Any further precise indication on proper citations are welcome. Authors: Printer-friendly version

Referee #1: Section 5.3. The potential of pollen for qualitative climatic reconstruction. I wonder if the authors can be more specific in discussing the analysis results by the data obtained because only abstract conception was mentioned here. The studies in Nakazawa and Fujita (2006, Annals of Glaciology) and Nakazawa et al. (2015, Environmental Earth Sciences) may be useful for the discussion in this chapter.

Authors: We note that data are discussed based on fig 4. This is only an example of how a climatic interpretation of the ice core, based on pollen dating, can be performed. More studies coupling high resolution pollen analyses on longer cores and measured meteorological data series are needed to provide further discussion points. Changes in the manuscripts: We nevertheless expanded the discussion using and citing the suggested papers.

Referee #1: Section 5.4. Application of the pollen based timescale. The authors need to clarify more the point of argument in this section. As the authors noticed, a good correlation between the mean daily temperature and the measured isotopic composition arises from preservation of seasonal variation of the D values. Therefore, the effect of re-evaporation or the stable isotope amount effect seems to be small. However, to reconstruct past temperature, the authors need to analyze the data while considering the smoothing of D values.

Authors: This is certainly a good point. However, this paragraph merely suggests

a potential use of the pollen timescale and does not suggest any further refinement of the basic interpretation of the dD record and we do not claim any more precise past temperature reconstruction. This topic is therefore not considered within the main scope of this publication.

Referee #1: Section 6. Conclusions. P9 L11: The timing of local flowering of different plant taxa and of the daily changes in airborne pollen concentration should be changed under climate change. How do the authors overcome this problem without airborne pollen data when applying this method to date deeper ice cores?

Authors: This is surely an important point that needs to be addressed when dating a deeper core. relevant. Studies on phenology and historic series of pollen monitoring encompassing the last 35 years show that the shift of the onset and end of the blooming season due to temperature trends are on average around 10 days in Europe (Menzel & Fabian 1999, Nature) as well as in the Ortles region (Bortenschalger & Borthenschlager 2007, Grana). We therefore assume that this difference is not particularly relevant for the dating of a deep core as this value is surely smaller than the number of days encompassed in one deep core sample. Changes in the manuscripts: We now address and explained this concept in the conclusion section.

Referee #1: Table 1. The dates in 2005 and 2006 are manifestly inconsistent with the EISModel calculation and the authors' arguments. It needs to be explain more.

Authors: We do not understand this comment as in Table 1 there are no Esimodel calculation. In our opinion Fig 3 shows, both visually and statistically, that 2005 and 2006 are quite consistent. This is also proven by the high correlation coefficient.

---

## Author Comment (AC2) · 14 Feb 2017

Authors: We thank the reviewer for her/his comments and the time dedicated to our manuscript.

Referee #2: Dating of ice cores are challenging subject especially in non polar ice caps, where melting can influence the signal. Here the authors aim at using pollen to date a shallow firn core from the South Tyrol alps with a day to day resolution by comparing results from ice cores with nearby station data. It is an interesting approach to use pollen to date ice cores. In a previous paper (Festi, 2015) the authors used the

same core and the same pollen data to date the record using PC and PCA methods to compare with airborne pollen samples from Solda. In this paper they use Jaccard similarities with the same airborne samples from Solda. They also in this paper use their highly resolved record to derive accumulation rates and compare those with a mass balance model, which as input use meteorological station data and move on to judge whether melt layers influence the pollen record.The main argument is based purely on the statistics, and the authors should consider how the sample depth may relate to the order of the samples. There is quite some indication of inverse orders, which to my opinion is not very well justified.

Authors, Changes in the manuscript: We now address specifically in sections 4.1 and 5.1, the inversions, clarifying their possible origin (wind erosion and redisposition, percolation, statistical artifact) and how this information contributes to the timescale.

Referee #2: However if accepting this kind of uncertainty in the dating of the samples, the pollen only arrive in spring/summer making the date of year dating only possible in spring/summer. This is not very clearly stated either.

Authors, Changes in manuscript: We now emphasize in paragraph 3.1. that the method can be applied in the spring and summer time since during winter and autumn there is no relevant pollen production in this region.

Referee #2 To me it is unclear that this paper brings much novelty to the already published paper by Festi (2015) in Journal of Glaciology.

Authors: The manuscript brings a very significant novelty in several respects :i) by presenting a new method to obtain a higher (sub-seasonal) resolution timescale based on pollen analyses; ii) by combining for the first time with highly interdisciplinary approach a pollen timescale with mass balance model (Eismodel); iii) by establishing the bases for a qualitative climate reconstruction of the ice cores dated by pollen. The manuscript includes a dedicated paragraph for each of these points. Changes in manuscript: We added a discussion section comparing our new results with those of Festi et al 2015

to show the coherence and the improvement of the timescale: from seasonality to subseasonality.

Referee #2 however I would suggest the authors to largely rewrite the manuscript and focus more on the novel aspects (as compared to the Festi (2015) publication, eg. the comparison to the accumulation model and the water isotopes as well as to emphasize the uncertainties and justify the inversions of the dating better.

Authors: We add a new discussion paragraph outlining the improvements obtained in the timescale precision with the day-to-depth method in comparison to the principal components (PCs) method (Festi et al 2015). In this paragraph, we provide evidence that the new method considerably improves the timescale based on the PCs listing the critical points of the PC method and indicating how the new method overcomes them. Furthermore, we now better address and support uncertainties and accuracy for example by comparing Eismodel calculation with mass balance observations carried out in the period from 2009 to 2013 at the study site. We also better discuss inversions and emphasise their origin and implication for the timescale. Changes in manuscript: We added a new discussion paragraph (Timescale improvements). As suggested, we expanded in section 4.1 and 5.1 the parts regarding the inversions considering how the sample depth relates to the order of the samples as well as discussing the possible physical and statistical origin of the incongruence.

Referee #2: Section 3.1 Define the time of year in which you can make depth to day comparisons based on pollen.

Authors: It was already reported in the methods (March to October) but now it has been further emphasised and more clearly stated as a limit of the method. Changes in the manuscript: P4L1 According to the suggestion we stressed the time-window of the year for which it is possible to apply the method.

Referee #2: Section 4.1, line 6 expand the explanation about inversions. And consider also in Table 1 to explain this inversions, as the table otherwise is very confusing with

dates going back and forth.

Authors: Thanks for the remark. In fact, inversions are quite interesting even if it is very hard to identify their origin. We decided to keep them and not to apply any adjustment because they can provide information about potential disturbance (wind redistribution and/or surface melt, as discussed in section 5.1) in the sequence. Changes in the manuscript: As suggested by the reviewer we added an explanation of the reason why we keep the inversions in section 4.1. Furthermore, a more detailed discussion of their origin and implications for the core chronology has been added in the discussion section 5.1.In the caption of table 1 we specify that the samples are listed in order of depth to avoid possible confusion, due to the inversions.

Referee #2: Section 4.1 in general should be more concise, it is very long and mainly lists the information from table 1.

Authors, Changes in the manuscript: We now reduced this section reporting synthetically the results of the pollen based timescale.

Referee #2: Section 4.1 eg. line 24, 31, 36 with more. The samples are 10 cm thick (?), how can layers be given in precision of cm, eg. 87 cm, 91 cm etc. These numbers should have uncertainties based on the sample sizes.

Authors: The depth is given in water equivalent and this is why depth are not merely 10-20-30 etc. We added a sentence also explaining that this is the "depth down" of the samples, meaning the value of the bottom depth of the sample.

Discussion paper Referee #2: Section 5.1 line 35/ Figure 3 (correlation figure). I am confused as it looks like you have more data in the correlation part of Figure 3 (eg. points during winter and autumn), where no points exists in the main part of figure 3 for pollen, where do these additional data stem from?

Authors: The graph contained also the winter and autumn points obtained by linear regression (paragraph 5.4 application of the pollen based timescale). We see how this

is confusing and therefore we changed it using only the spring and summer dates. Thanks for the remark. Changes in the manuscript: The graph has been updated as suggested.

Referee #2: Section 5.2 Discusses melt layers, however nowhere in any figures are the position of melt layers shown. Please add melt layers to relevant figures, eg. as vertical bars in Fig 2 (and Fig 3).

Authors, Changes in the manuscript: We now added melt layers as required in Fig 3 and ice lenses stratigraphy has been added to Fig 2.

Referee #2: Section 5.4 (figure 5) . Here you discuss the comparison to water isotopes. In the years 2007, 2008 and 2009 the water isotope data from april to July gets very steep, followed by a somewhat not steep slope down to the next winter. This seems like an effect of you having the pollen data very specifically dated in exactly those months. You suggest that those years are fine, but rather 2006 and 2005 are influenced by meltwater percolation from summer into the winter affecting the summer peak. This may very well be, but it does not explain the lack of similarities for the later years, where the steep slope to me looks very artificial. I would suggest you add some comments about the uncertainty of the dating, it could eg. be explained by later blooming of the pollen shifting the summer bloom and extending the dD peaks to have a more sinusoidal behavior as also observed in the Solda record of temperature.

Authors: We understand the reviewer's impression, however the steep slope in in the years 2007 and 2008 is due to the abundant snow accumulation in the springtime as it can be seen in Fig 3. The simulation of snow accumulation by Eismodel (Fig 3) points to a rapid increase in the snow cover resulting in the fact that several samples have similar deposition dates and cluster in a steep slope also in Fig 3.

---

## Author Response (AR1)

Dear Editor and Reviewers,

Please find the marked up version of the revised manuscript here below.

I would like to draw your attention to the fact that the numbering of the discussion paragraph in the reviewed version of the manuscript changed due to the addition of a discussion paragraph entitled "Improvements in the pollen timescale" and numbered as 5.1. The order of the discussion paragraphs included in the previews version remains the same but the numbering changed, i.e. paragraph 5.1 is now 5-2 etc.

Thank you for your attention.

Daniela Festi & coauthors

[revised manuscript text omitted]

---

## Editor Decision (ED1)

| Page | Line | Comment |
|---|---|---|
| 2 | 36-39 | Drop these lines. This remains a weakness of the paper. I would like you to delete it. See further |
| 5 | 12 and 13 | ..and in related figure: I would choose another color of these inversions in nthe graph to make these pop up clearly |
| 6 | 25 | delete "." at end of sentence |
| 7 | 2 | blank missing before "provide" |
| | 4 | change to: "the new method shows clear improvments at various levels" |
| | 8 | Change to: "For example, in the year 2005…" |
| | 16 | delete "etc." |
| | 31 | replace "pollen dating" by "pollen analysis" |
| | 34 | delete sign before "This" |
| | 36 | delete sign before "the modelling" |
| | 36 and 37 | Change to: "the modelling approach appears globally robust following validation…" |
| | 37 to 39 | This is important. It deserves another insert plot in Figure 3, of Eismodel vs. Field validation |
| 8 | 3 | Change to: "Such efficiency, however, cannot.." |
| | 4 and 5 | This is making the "depth-to-date" application more problematic then.. I think it should be mentioned |
| | 11 and 12 | change to: "helps in in the interpretation…inversions *(specific* color dots in Figure 2b). |
| | 13 | Change to: " can potentially be of two origins: a) there might be…artefact.." |
| | 15 | Change to: "accuracy; b) on the other hand, " |
| | 17 | Change to: ".. and snow: redistribution of..., and mixing…" |
| | 20 | Change to: "..in section 5.3" |
| | 36 | There are discordances between the depths of "ice lenses" in figure 2d and "meltlayers" in Figure 3! Needs consistency! |
| | 41 | delete "completely" |
| 9 | 7 | Also discordances between depths of "ice lenses" and "meltlayers" Figure 2 and 3 |
| | 7 | delete "-" in front of "A further.." |
| | 14 | change to: "..where these authors.." |
| | 19 | Change to: "However, in the same study, no evidnece.." |
| | 23 | delete "-" in front of "These" |
| | 27 | Change to: "..2010) similar values being reported.." |
| | 28 | Change to: "could show…" |
| | 29 | replace "dislocation" by "transport" |
| 10 | 3 | spelling: "recognized" |
| | 22 | change to: "..in order to support it." |
| | 36 and 37 | But then, if that is the case, one can wonder how far down in the deep ice core you will be able to decipher "sub-seasonal" behaviour?...Also, changes between glacials and interglacials are likely to involve more drastic changes than some 10 days of shift in the flowering timing!.. The whole speciation could change!...This is not convincing and I would delete lines 32 to 37... I would also add in the previous sentence that the feasibility of the approach in deep ice cores also relies on potential regional shifts in pollen speciation. |
| 10 and 11 | All 5.5 | This remains a very weak section of the paper!.. The T°/deltaD relationship is really blurred and has nothing to do with what would be expected of a clean GMWL!.. It is therefore very difficult to interprete, and, to me, brings no added value. It is simply pushing your application too far!.. That section is not convincing and should be dropped. |
| 11 | 23 | Change to: "..source, with negligible source-sink lag, the existence.." |
| | 29 | Change to: "…and can also take place …Alternatively, data from the closest.." |
| | 32 | Change to: ".. As witnessed by the recent launch…" |
| | 33 | Change to: "..Theoretically, our approach could be applied to deeper ice cores, if the sampling resolution…high and if no significant shift occurred in the pollen specification. In such cases our method could … model by documenting the loss of seasonal signal with depth in the core." |
| | 41 | replace: "types" by "conditions" |
| 12 | 1 to 3 | Too weak. Drop this. |
| | 8 | Change to: ".. In this paleo-climatic archive under "borderline" environmental conditions for proxies reliability." |
| Figure 2 | caption | "Number and thickness of ice lenses": it is not clear if this is total thickness or individual thickness.. Surely not all ice lenses at a given depth have the same thickness (?) |
| Figure 4 | axes | Change to: "Mass balance from modelling" and "Mass balance from Pollen" |
| Figure 5 | | Delete |

---

## Author Response (AR2)

Innsbruck, 21/03/2017

Dear Jean-Luis Tison,

Thank you for your useful comments and suggestions. I made all the changes required (see list of changes attached) and I also decided to drop the section 5.5. as suggested. As a consequence also the title and the abstract changed slightly (see marked up version attached below).

Finally, I would like to thank you for your patience and efforts as Editor and for giving us the possibility to publish our work in The Cryosphere.

I wish you all the best for your trip to Antarctica!

Sincerely,
Daniela Festi

**LIST OF CHANGES**

Editors comments in red.
Authors reply in black.

**PAGE 2**

L236-39: Drop these lines. This remains a weakness of the paper. I would like you to delete it. See further512

Authors: Done.

**PAGE 5**

L12-13..and in related figure: I would choose another color of these inversions in the graph to make these pop up clearly

Authors: Done. We used empty squares. Fig. capture has been updated as well.

**PAGE 6**

L25: delete "." at end of sentence

Authors: Done.

**PAGE 7**

L 2: blank missing before "provide"

Authors: Done.

L 4: change to: "the new method shows clear improvements at various levels"

Authors: Done.

L8: Change to: "For example, in the year 2005…"

Authors: Done.

L16: delete "etc."

Authors: Done.

L31replace "pollen dating" by "pollen analysis"

Authors: Done.

L34 delete sign before "This"

Authors: Done.

L36 delete sign before "the modelling"

Authors: Done.

L36 and 37Change to: "the modelling approach appears globally robust following validation…"

Authors: Done.

L37 to 39This is important. It deserves another insert plot in Figure 3, of Eismodel vs. Field validation

Authors: Since it was very hard to insert a further graph into Fig 3 we added a Figure 4 containing the 2 scatter plots (the one previously included in Fig 3 (comparison EISmodel and pollen dates) and the one here required.

**PAGE 8**

L3: Change to: "Such efficiency, however, cannot.."

Authors: Done.

L4 and 5: This is making the "depth-to-date" application more problematic then.. I think it should be mentioned

Authors: We added a sentence. I agree that if there is such a delay it must be considered, quantified and integrated in the model.

L11 and 12: change to: "helps in in the interpretation…inversions (specific color dots in Figure 2b).
Authors: Done.
L 13: Change to: " can potentially be of two origins: a) there might be…artefact.."
Authors: Done.
L 15: Change to: "accuracy; b) on the other hand, "
Authors: Done.
L17: Change to: ".. and snow: redistribution of..., and mixing…"
Authors: Done.
L20: Change to: "..in section 5.3"
Authors: Done.
L36: There are discordances between the depths of "ice lenses" in figure 2d and "meltlayers" in Figure 3! Needs consistency!
Authors: They are not discordances. In temperate firn ice lenses are not expected to form only in correspondence of meltlayers but also as a result of meltwater percolating from the surface and refreezing over internal discontinuities of the snowpack, such as sun crusts or wind crusts.
L41: delete "completely"
Authors: Done.

**PAGE 9**
L7:Also discordances between depths of "ice lenses" and "meltlayers" Figure 2 and 3
Authors: See comment above. Also in this case the water seems to have percolated till the winter layers forming a thick ice lens.
L7:delete "-" in front of "A further.."
Authors: This seems to be a deleted blank. See now Page 7 L 29.
L14change to: "..where these authors.."
Authors: Done.
L19Change to: "However, in the same study, no evidnece.."
Authors: Done.
L23delete "-" in front of "These"
Authors: This was also a deleted blank. See now in P8 L4.
L27Change to: "..2010) similar values being reported.."
Authors: Done.
L28Change to: "could show…"
Authors: Done.
L29 replace "dislocation" by "transport"
Authors: Done.

**PAGE 10**
L3: spelling: "recognized"
Authors: Done.
L22:change to: "..in order to support it."
Authors: Done.
L36 and 37:But then, if that is the case, one can wonder how far down in the deep ice core you will be able to decipher "sub-seasonal" behaviour?...Also, changes between glacials and interglacials are likely to involve more drastic changes than some 10 days of shift in the flowering timing!.. The whole speciation could change!...This is not convincing and I would

delete lines 32 to 37... I would also add in the previous sentence that the feasibility of the approach in deep ice cores also relies on potential regional shifts in pollen speciation.

Authors: We deleted those lines and added the sentences suggested (and the corresponding citations from Reference lists).

**PAGE 10 and 11**

All 5.5 : This remains a very weak section of the paper!.. The T°/deltaD relationship is really blurred and has nothing to do with what would be expected of a clean GMWL!.. It is therefore very difficult to interprete, and, to me, brings no added value. It is simply pushing your application too far!.. That section is not convincing and should be dropped.

Authors: Deleted.

PAGE 11

L23: Change to: "..source, with negligible source-sink lag, the existence.."

Authors: Done.

L29: Change to: "…and can also take place …Alternatively, data from the closest.."

Authors: Done.

L32: Change to: ".. As witnessed by the recent launch…"

Authors: Done.

L33: Change to: "..Theoretically, our approach could be applied to deeper ice cores, if the sampling resolution…high and if no significant shift occurred in the pollen specification. In such cases our method could … model by documenting the loss of seasonal signal with depth in the core."

Authors: Done.

L41replace: "types" by "conditions"

Authors: Done.

**PAGE 12**

L1 to 3: Too weak. Drop this.

Authors: Done.

L8: Change to: ".. In this paleo-climatic archive under "borderline" environmental conditions for proxies reliability."

Authors: Done.

**FIGURES**

Figure 2, caption "Number and thickness of ice lenses": it is not clear if this is total thickness or individual thickness.. Surely not all ice lenses at a given depth have the same thickness (?)

Authors: it refers to mean individual thickness of lenses per sample (added in capture).

Figure 4: axes Change to: "Mass balance from modelling" and "Mass balance from Pollen"

Authors: Done. Fig. 4 has been renumbered as Figure 5.

Figure 5: Delete.

Authors: Former figure 5 has been deleted.

[revised manuscript text omitted]

---

## Author Response (AR3)

Innsbruck, 22/03/2017

Dear Jean-Lous Tison,
Editor of The Cryoshpere,

Thank you for your final suggestions. We have made all the changes requested to our manuscript. Please find attached the list of changes and the marked up version of the manuscript.

Kind regards,

Daniela Festi & coauthors
* * *
**LIST OF CHANGES:**

p. 6, line 30: Should be "Figure 3a" instead of "Figure 4"...to keep the sequence of the figures numbering as it is called in the text
Authors: Done.
p.6, line 32: should refer to 'Fig. 4" only
Authors: Done.
p.6, line 36: change to: "..August 2013 (Fig. 4b)"
Authors: Done.
p.6, line 39: should refer to : "..(Fig. 4 and 5).."
Authors: Done.
p.7, line 6: change to: "...inversions (empty squares in Fig. 2b)..."
Authors: Done.
p. 10, line 10: delete the second "also"
Authors: Done.
p. 10, line 13: change to: "witnessed"
Authors: Done.
p.10, line 16: change to "could" (d missing)
Authors: Done.
p.11, line 12: my first name is "Jean-Louis" :0) ..
Authors: Done. Sorry for misspelling!
Figure 3 becomes Figure 4 and vice versa
Authors: Done.
Caption of new Figure 3: should be "2008 to 2013" .. if we follow the text
Authors: Done.

[revised manuscript text omitted]